# Effective Removal of Metal ion and Organic Compounds by Non-Functionalized rGO

**DOI:** 10.3390/molecules28020649

**Published:** 2023-01-08

**Authors:** Viviana Sarmiento, Malcolm Lockett, Emigdia Guadalupe Sumbarda-Ramos, Oscar Vázquez-Mena

**Affiliations:** 1Facultad de Odontología, Universidad Autónoma de Baja California, Tijuana 22427, BC, Mexico; 2Department of NanoEngineering and Center for Memory and Recording Research, University of California San Diego, La Jolla, CA 92093, USA; 3Facultad de Ciencias de la Ingeniería y Tecnología (FCITEC), Universidad Autónoma de Baja California, Valle de las Palmas, Tijuana 22427, BC, Mexico

**Keywords:** reduced graphene oxide, lead ion removal, aqueous samples, metal removal

## Abstract

Effective removal of heavy metals from water is critical for environmental safety and public health. This work presents a reduced graphene oxide (rGO) obtained simply by using gallic acid and sodium ascorbate, without any high thermal process or complex functionalization, for effective removal of heavy metals. FTIR and Raman analysis show the effective conversion of graphene oxide (GO) into rGO and a large presence of defects in rGO. Nitrogen adsorption isotherms show a specific surface area of 83.5 m^2^/g. We also measure the zeta-potential of the material showing a value of −52 mV, which is lower compared to the −32 mV of GO. We use our rGO to test adsorption of several ion metals (Ag (I), Cu (II), Fe (II), Mn (II), and Pb(II)), and two organic contaminants, methylene blue and hydroquinone. In general, our rGO shows strong adsorption capacity of metals and methylene blue, with adsorption capacity of q_max_ = 243.9 mg/g for Pb(II), which is higher than several previous reports on non-functionalized rGO. Our adsorption capacity is still lower compared to functionalized graphene oxide compounds, such as chitosan, but at the expense of more complex synthesis. To prove the effectiveness of our rGO, we show cleaning of waste water from a paper photography processing operation that contains large residual amounts of hydroquinone, sulfites, and AgBr. We achieve 100% contaminants removal for 20% contaminant concentration and 63% removal for 60% contaminant concentration. Our work shows that our simple synthesis of rGO can be a simple and low-cost route to clean residual waters, especially in disadvantaged communities with low economical resources and limited manufacturing infrastructure.

## 1. Introduction

Water pollution is one of the most important and challenging problems in our society that may have disastrous consequences [1]. Industrialization has brough as a side effect dangerous residues, such as heavy metals, organic dyes, and organic bleachers, that quite often contaminate the environment and the water for human use. Ingestion of heavy metals through drinking water can have serious effects on human health [2,3,4,5,6,7,8,9]. Heavy metals can affect respiratory [10], circulatory [11,12,13], and renal systems [14,15,16], and even produce malformation in fetuses [17,18]. Similarly, organic dyes, such as methylene blue (MB) and phenol-based bleachers like hydroquinone (HQ), have shown toxic effects on human health [19,20,21,22,23,24,25]. The direct consequences for dye effluents include depletion of dissolved oxygen levels, decreased reoxygenation potential, leaching of dyestuff from soil into groundwater, and reduced light penetration into water for photosynthesis [26,27,28,29,30]. In addition, underserved communities with low economical resources tend to be more vulnerable due to poor infrastructure. Therefore, it is important to develop low-cost approaches that balance cost and effectiveness to remove heavy metals and organic contaminants from residual waters in order to enable its proper handling, either for safe waste disposal or for safe reutilization for human use. Several techniques exist to remove contaminants [31], such as filtration [32,33], reverse osmosis [34], precipitation [35], and adsorption [36,37,38,39,40], among others, including the use of clays [41,42,43,44,45,46]. The large surface-to-volume ratio of nanoscale structures has attracted a lot of attention for filtering and adsorption applications, such as heavy metal removal [47,48,49,50]. In particular, graphene based-materials have large surface-to-volume ratio [51,52], strong chemical [53] and mechanical [54,55,56,57] stability, and potential for large and low-cost production [58,59,60], making them excellent candidates for adsorbtion of ion metals [31,37,38,50,61,62,63,64,65,66,67,68]. An important strategy for ion metals removal has been the functionalization of graphene oxide (GO) derivatives with oxygen, sulfur, and nitrogen groups to improve its affinity for heavy metals [62,65]. Even when using reduced graphene oxide (rGO), the functionalization of basal planes has been an important strategy to improve the adsorption of heavy metals and organic contaminants [69,70], with fewer reports on bare reduced graphene oxide as adsorbing material [71,72,73]. However, an alternative path is the use of defects in graphene, such as vacancies that can trap metal atoms, as shown from first-principle density-functional studies [74,75,76,77], avoiding the addition of any functional groups that may add some complexity or cost to the material preparation.

Herein, we demonstrate the easy and low-cost wet synthesis of rGO using gallic acid (Gall) and sodium ascorbate obtained at 200 °C and without any functionalization. This method induces several defects in the basal planes that serve as anchoring points to trap metal atoms and organic contaminants, showing high adsorption capacity with respect to previous reports on non-functionalized rGO materials. We present the characterization of the rGO, studying nitrogen adsorption and desorption isotherms to characterize the specific surface area and pore diameter. We perform FTIR and Raman spectroscopy to study the conversion from GO to rGO, as well as zeta-potential analysis to estimate the reactivity of the material. Then, we study the adsorption process for Ag (I), Cu (II), Fe (II), Mn (II), and Pb (II), as well as for methylene blue (MB) and hydroquinone (HQ). For all these compounds, we study the adsorption dependence on pH conditions and on initial concentrations to extract maximum adsorption capacities exploring Langmuir and Freundlich models. Finally, to prove the effective application of the material, we treat wastewater from paper photography processing operations using rGO, showing efficient removal of contaminants.

## 2. Results and Discussion

### 2.1. rGO Properties

The conversion of GO into rGO using gallic acid and sodium ascorbate, and the adsorption interactions of metal ions with rGO, are shown in Figure 1A. The rGO was analyzed by FTIR and Raman spectroscopy. The FTIR spectra for the GO and rGO are shown in Figure 1B. The GO shows the characteristic absorption peaks from sp^2^ carbon bonds and oxygen functional groups located at 3350, 1720, 1620, and 1390 cm^−1^, which correspond to O-H, C=O, C=C, and C-O-C stretching vibration modes, respectively [78]. The bands at 1170–1040 cm^−1^ are attributed to C–O stretching vibrations, confirming the existence of phenols. After the reduction process, significant reduction in the absorption of the oxygen functional groups peaks is observed. The O-H peak (3350 cm^−1^) practically disappears, while the C=O and C=C peaks (1720 and 1620 cm^−1^) are strongly reduced but still present. The existing C=C peak at 1620 cm^−1^ in the spectra of rGO samples suggests that the sp^2^ structure of carbon atoms remains as expected. Moreover, the relative decrease in the intensity at the 1720 cm^−1^ and 1170 cm^−1^ peaks indicates that C=O and C-O stretching of the carboxylic acid groups and epoxy groups, respectively, still exists, but in lower proportion. [78] This is expected because many functional groups remain attached to the basal planes, preventing a strong interaction between graphene basal planes and allowing molecules to penetrate the interlayer space for adsorption applications. Raman spectroscopy shown in Figure 1C also confirmed the conversion from GO to rGO. The GO spectrum shows the D peak at 1365 cm^−1^, which appears in disordered graphene structures [59,60], as well as the carbon sp^2^ bonding signature G peak at 1595 cm^−1^. After the reduction, there are no significant shifts in the peak position. However, the D and G peak ratio (I_D_/I_G_) changes, increasing from 0.936 for GO to 1.287 for rGO, which has been previously reported for graphene reduction with gallic acid, hence, confirming the reduction process. [79,80,81].

A Brunauer–Emmett–Teller theory analysis was performed to estimate the specific surface area of the rGO. Figure 1D shows the nitrogen (N_2_) adsorption-desorption isotherm carried at 77.25 K, exhibiting a type IV isotherm with a hysteresis loop above P/Po = 0.9 that indicates the mesoporous nature of the material [82]. The extracted specific surface area obtained was 83.5 m^2^/g. This is lower compared to the hydrothermal rGO aerogels reaching 136 m^2^/g [73] and 1000 m^2^/g, incorporating mesoporous silica [83], but larger than aerogels incorporating TiO_2_ (74.9 m^2^/g) [84], carbon nanofibers (38.9 m^2^/g) [85], and Fe_3_O_4_ magnetic nanoparticles(58 m^2^/g) [86]. Such high specific surface area provides more active adsorption sites for the electrostatic reaction, leading to an enhanced adsorption uptake [87]. The Barrett–Joyner–Halenda method was used to extract the pore size of the rGO. Figure 1E shows the corresponding pore size distribution plot calculated from the N_2_ adsorption isotherm and the Barrett–Joyner–Halenda model. The plot shows a pore size distribution with a mean size in the ~14.06 nm range, confirming the mesoporous nature [82,88] of the rGO and its suitability for metal adsorption. Information on the presence of binding sites in the rGO in the form of surface charges was obtained by measuring the zeta potential of GO and rGO. Figure 1F shows the zeta potential values of GO and rGO, which were −32.2 ± 1.54 and −52.5 ± 1.20 mV, respectively. The presence of large negative charges favors the trapping of positive ions, such as metallic or organic pollutants. From zeta potential measurements, it is possible to estimate the flake size, as shown in Figure 1G. The average size distributions of GO and rGO were 2057.3 ± 163.3 and 529.37 ± 38.7 nm, respectively. The smaller size for rGO could be due to the ripping process of the basal planes, when oxygen-function groups are removed from the graphene basal planes, leading to smaller flakes with more charged/reactive sites.

### 2.2. Adsorption of Metal Ions

The pH of an aqueous solution is an important variable that affects the adsorption of ions at solid-water interfaces. We keep our measurements at pH < 8 because higher pH values result in ion precipitation. [89,90,91] The adsorption capacity (*q_e_*) by rGO of the Ag (I), Cu (II), Fe (II), Mn (II), and Pb (II) metal ions as a function of pH is plotted in Figure 2A. The adsorption has a similar behavior for all the tested metals, increasing from pH 3 until reaching the maximum value at pH 6, followed by a decline towards pH 8. At low pH values, the large number of H^+^ can compete with the metal ions for binding sites in the rGO, leading to lower *q_e_* for all the metals. By increasing pH value, the concentration of H^+^ decrease and the tendency of the metal ion to occupy active sites increases, leading to an increase of *q_e_* of rGO. When the pH is higher than the optimum pH (in our study pH_optimal_ = 6), the metal ions may get converted to their hydroxides, resulting in a decrease in the removal of metals by the active sites of rGO. [92,93,94,95] For Pb (II), at pH > 6, *q_e_* is reduced because the predominant metal ions are Pb(OH)^−^_3_ that cannot be adsorbed on the surface of rGO. [96] These results also show that among the metals tested, the highest adsorption capacity is for Pb (II) through all pH conditions, while Mn (II) and Cu (II) show the lowest adsorption capacities. For Pb (II), *q_e_* increases from 140 mg/g at pH = 4, reaching 240 mg/g at pH = 6, and then dropping drastically to 100 mg/g at pH = 8. For Cu (II), one of the lowest adsorbing materials, *q*_e_ = 25 mg/g at pH = 2, increasing to its maximum *q*_e_ = 55 mg/g at pH = 6, and then dropping sharply to *q*_e_ = 15 mg/g at pH = 8. To estimate the maximum adsorption capacity, the conditions for the following experiments were kept at pH = 6.

In order to test the ion metal adsorption capabilities of our rGO, we performed single metal-ion equilibrium adsorption measurements for Ag (I), Cu (II), Fe (II), Mn (II), and Pb (II). Figure 2B shows the extraction percentage as a function of the initial concentration in an aqueous solution. As expected from the pH measurements, Pb shows the highest adsorption capacity, with Cu, Fe, Ag, and Mn showing lower adsorption capacity. Pb (II) extraction reaches 92% at 100 mg/L initial concentration, Cu (II) reaches 93% extraction at 25 mg/L, Fe (II) reaches 97.14% extraction at 25 mg/L, Ag (I) reaches 92.52% extraction at 25 mg/L, and Mn (II) reaches 99% extraction at 50 mg/L. To study the adsorption capacity of the rGO, we used the Langmuir model (Equation (4)) because several graphene-based adsorbing materials show a monolayer adsorption behavior. Figure 2C shows the behavior of $q_{e}$ as a function of the equilibrium concentration *C_e_*, and Figure 2D shows the linear fitting to the Langmuir model of *q_e_/C_e_* as a function of *C_e_*, allowing extraction of *q*_max_ and *K*_L_. The fitting parameter R^2^ is >0.99 for all the metals, indicating the good fitting with the Langmuir model and, therefore, indicating a monolayer coverage as expected. The values of *q*_max_ and *K*_L_ are listed in Table 1 for the metals tested. The values obtained are *q*_max_ = 243.9 mg/g for Pb(II), *q*_max_ = 80.64 mg/g for Fe(II), *q*_max_ = 78.12 mg/g for Cu(II), *q*_max_ = 63.29 mg/g for Ag(I), and *q*_max_ = 57.80 mg/g for Mn(II). These results confirm that Pb(II) shows the best adsorption, followed by Fe, Cu, Ag, and Mn.

Table 2 compares the performance of our rGO adsorbing metal ions with other literature reports. Our adsorption results are very competitive compared with other rGO structures. For example, for Pb(II), our *q*_max_ value of 243.9 mg/g is higher than results for rGO-F_3_O_4_ nanocomposites with *q*_max_~30–50.00 mg/g [86,97], nickel ferrite-rGO with *q*_max_~120−150 mg/g [98,99], rGO produced from algal extracts with *q*_max_~95 mg/g [100], and porous rGO aerogel produced by the hydrothermal method with *q*_max_~58.04 mg/g [73]. Our results for Pb (II) and Cu (II) are also better than those reported for 2-imino-4-thiobiuret- partially reduced graphene oxide, reporting *q*_max_~102.2 mg/g for Pb(II) and *q*_max_~37.9 mg/g for Cu(II) [70]. Compared with rGO prepared by reduction with ethylenediamine, we have mixed results; their performance for Cu (II) (*q*_max_~55.34 mg/g) and Mn(II) (*q*_max_~42.46 mg/g) is lower than our rGO, but for Pb, they have significantly superior performance with (*q*_max_~413.34 mg/g) [71]. Our performance is still significantly lower than functionalized GO-based materials that can reach *q*_max_~450 mg/g for Pb (II) and Cu (II) using chitosan/GO nanofibers [101]. Given that our material has its optimal performance close to neutral pH and that our synthesis does not involve any heating or subsequential functionalization, we believe it is a strong candidate for effective and low-cost heavy metal removal, offering a very good compromise between adsorption capacity and simple synthesis.

### 2.3. Adsorption of Methylene Blue (MB) and Hydroquinone (HQ)

For MB and HQ, pH also plays an important role, as shown in Figure 3A, with a maximum removal at pH = 6 for both substances. The removal of MB increases drastically when the pH increases from 2 to 6, following by a decrease from pH = 6 to 8. At low pH, H^+^ groups compete with dye cations, decreasing the amount of dye adsorbed [103]. As the pH increases, the negative charge of the functional groups on rGO (Figure 1B) increase dye adsorption due to electrostatic attraction between positively charged sorbate and negatively charged sorbent [104]. For HQ, the pH affects the protonation equilibrium of ligands in solution and the protonation level of individual surface sites [105]. At low pH (pH < 6), the hydroquinone is mainly presented as a neutral molecular form, which is adsorbed by rGO. At higher pH (pH > 6), the OH^−^ competes with deprotonated ligand for the adsorption sites on rGO [106]. The negatively charged phenoxy ions, such as hydroquinate and hydroquinate dianions, increase gradually, whereas the functional groups of rGO (Figure 1B) become negative, leading to electrostatic repulsion and a decrease in the removal efficiency at high pH [107]. It is probable that the strong π–π interaction between the protonated form of HQ, as well as the cation–π bonding between the protonated amino group or sulfur atom of MB and the π-electron-rich aromatic structure of GO, may play important roles in their binding to GO.

The adsorption amount of MB and HQ is effectively dependent on the initial concentration of the compounds. Figure 3B shows how material extraction (%) of MB and HQ decreases as the initial concentration increases, as expected. However, the adsorption for HQ shows a distinctive behavior, with a fast decay at low initial concentrations (C_0_ = 10–40 mg/L), followed by a slower decay at larger concentrations (C_0_ = 50–100 mg/L). Figure 3C shows q_e_ vs. C_e_ for MB and HQ with a Langmuir fitting, showing a good fit for MB. However, for HQ, there is some deviation for large concentrations. Using data from Figure 3C, we obtained both the Langmuir and Freundlich isotherm fitting parameters that are shown in Table 3. In the case of MB, Langmuir shows a slightly better fitting (R^2^ = 0.9878) than the Freundlich model (R^2^ = 0.9737), indicating that a monolayer coverage seems more adequate to describe the MB behavior with a *q*_max_ = 238.45 mg/g. The Langmuir fitting for MB is shown in Figure 3D. For HQ, given the different decays for low and high concentrations observed in Figure 3B, we performed Langmuir and Freundlich isotherm fittings for low (HQ_L_) and high (HQ_H_) concentrations. The results are shown in Table 3, demonstrating that for a low concentration, the Langmuir fitting gives better fitting (Langmuir: R^2^ = 0.9937, Freundlich: R^2^ = 0.9780), thus indicating a monolayer process, as expected, with *q*_max_ = 51.02 mg/g. However, for high concentrations, the Freundlich model gives a better fitting (Langmuir: R^2^ = 0.7235, Freundlich: R^2^ = 0.9923), suggesting a multilayer process with adsorption capacity of K_f_ = 0.834. The fitting plots, Langmuir for HQ_L_ and Freundlich for HQ_H_, are shown in Figure 3E,F. Further microscopic studies are required to understand the nature of the adsorption of HQ in order to explain the transition from a Langmuir to a Freundlich model.

Table 4 shows previous reports on MB adsorption using graphene-based materials to compare with our results, indicating that our rGO has a strong performance at *q*_max_ = 238.45 mg/g, especially considering the simplicity of our rGO synthesis. Our adsorption capacity is higher than several previous reports, including for graphene nanoplatelets with *q*_max_ = 225 mg/g [108] or graphene oxide/calcium alginate composites [109] (*q*_max_ = 181.81 mg/g), among others. However, our performance is lower than those of GO, with high degrees of oxidation that achieved two to seven times greater adsorbent capacity than our material [110,111]. The comparison with other reports for HQ adsorption are shown in Table 5. As previous results report Langmuir fittings, for the purpose of comparison, we use a conservative value of *q*_max_ of ~150 mg/g for HQ obtained from Figure 3A at pH = 6. Our results show greater adsorption capacity than organobentonites [112] (*q*_max_ = 12.05–21.55 mg/g), Fe granular-activated carbon (*q*_max_ = 26.65 mg/g) [113], Graphene aerogel-mesoporous silica hybrid (*q*_max_ = 67 mg/g) [83], and granular activated carbon (*q*_max_ = 102−135 mg/g) [114]. However, our results are below those obtained by *Phragmites australis*-activated carbon (*q*_max_ = 156 mg/g) [115], amino-poly (Vinylamine)-functionalized GO-(o-MWCNTs) Magnetic Nanohybrids (*q*_max_ = 293 mg/g) [116], and Graphene Oxide functionalized with Magnetic Cyclodextrin–Chitosan (*q*_max_ = 428 mg/g) [117]. Moreover, our results show that our rGO presents a competitive performance capturing MB and HQ compared with non-functionalized graphene materials, especially when considering the simplicity and ease of its synthesis in our method.

### 2.4. Cleaning Waste Water from Paper Photography Processing Operation

Finally, as a practical test, we tested the effectiveness of our rGO to clean the waste water from paper photography processing operations (WPO). The residues from processing negatives and photography paper contain high levels of hydroquinone, sulfites, and AgBr. To study the adsorption behavior, we tested different dilution rates to analyze different concentrations of contaminants. Figure 4A shows a visual comparison of WPO treated with rGO for 48 h with respect to WPO without rGO treatment as a control sample. The photographs show the change in color after the adsorption for the sample with rGO, indicating the removal of contaminants in the wastewater. The rGO with adsorbed contaminants precipitates after the adsorption process. To analyze the adsorption performance, total dissolved solids (TDS) measurements were measured in the WPO sample before and after treatment with rGO. TDS includes organic and inorganic salts that are completely dissolved in the water. Figure 4B shows that for WPO diluted to 20% with clean water, we have nearly total 100% extraction of the compounds present in the WPO, whereas for higher dilutions of 60%, the material removal decreased to 63%. In parallel, we also studied the rGO before and after the adsorption tests. Figure 4C shows a Raman analysis of the rGO before and after, demonstrating in both cases the expected high D and G peaks characteristic of graphene, and that the structure of the graphene remains unaltered after the adsorption events. However, FTIR in Figure 4D shows a clear change before and after adsorption. Before adsorption, rGO shows a relatively flat behavior, similar to the behavior of Figure 1B after reduction of GO. However, after the adsorption process, several peaks appear in the FTIR of rGO in the 3000–3500 cm^−1^ and 500–1000 cm^−1^ ranges, indicating binding events probably associated with adsorbing contaminants from the solution. Figure 4B,D thus show the effectiveness of rGO removing contaminants from WPO. Some pre-dilutions or multiple cycles may be required to have high effectiveness, but the technique is cost effective given the simplicity of rGO synthesis.

## 3. Experimental Section

*Chemical Reagents.* Graphene oxide was purchased from Graphenea (Spain). Analytical reagent grade Gallic acid (Gall), sodium ascorbate (NaLAc), NaOH, HNO_3_ were purchased from Sigma Aldrich. Dimethylformamide (DMF), hydroquinone, and methanol were purchased from Thermo Fisher Scientific. CuCl_2_·2H_2_O, Pb(NO_3_)_2_, MnCl_2_·4H_2_O, AgNO_3_, FeCl_2_·4H_2_O were purchased from Merck Millipore. Methylene blue lab grade powder was purchased from Innovating Science.

*Preparation of reduced graphene oxide (rGO).* Gallic acid (1 g) and sodium ascorbate (1 g) were added to the mixture of 20 mL deionized water and 20 mL DMF. The mixture was heated to 50 °C until the gallic acid and sodium ascorbate dissolved fully. Then, 0.25 g Graphene oxide (GO) was added into this system, ultrasonically dispersed for 30 min, and heated to 200 °C for 3 h. This mixture was stirred at room temperature for 24 h. The reaction liquid was filtrated and washed with methanol and deionized water several times. The product was dried fully at 60 °C, obtaining reduced Graphene Oxide (rGO).

*rGO characterization.* FT-IR measurement was performed on a Bruker VERTEX70 infrared analyzer. Raman spectroscopy was conducted on a Renishaw Invia Raman microscope spectrometer. Samples were excited with a 532 nm green laser and 50× objective lens. To avoid burning the samples, the power source was set to 1%. BET Analysis based on N_2_ adsorption and desorption isotherms of rGO were determined by an ASAP2020 HD88 instrument. Zeta potential was performed using a Metrohm 910 PSTAT potentiostat.

*Batch Mode Adsorption Experiment.* Stock solutions of various concentration of CuCl_2_·2H_2_O, Pb(NO_3_)_2_, MnCl_2_·4H_2_O, AgNO_3_, and FeCl_2_·4H_2_O were used as sources for Cu (II), Pb (II), Mn (II), Ag (I), and Fe (II) ions, respectively. The adsorption capabilities of rGO for organic compounds were assessed using methylene blue and hydroquinone as adsorbates. The experiments for batch mode adsorptions were conducted at room temperature (RT) to study the effect of pH, rGO dosage, and contaminant adsorbate concentration. In a typical experiment, different amounts of rGO were added separately to 20 mL of a concentration of adsorbate in water. The mixtures were stirred at room temperature for 48 h to achieve adsorption equilibrium, then they were centrifuged (TFCL electric lab Benchtop Centrifuge) at 4000 rpm for 5 min, and finally the supernatant was used for analysis. A Hanna Instruments Multiparameter Photometer, Serie C99, was used for the determination of Ag (I), Cu (II), Fe (II), and Mn (II) concentrations in the solution before and after adsorption by the rGO. For Pb (II) determination, we used a DR3900 Benchtop VIS Spectrophotometer (RFID Technology, Hach company). The residual concentrations of Methylene blue and hydroquinone, were measured using a UV–Vis spectrophotometer (PerkinElmer Lambda 35) at the corresponding maximum absorption wavelength (λmax: Methylene blue 664 nm, Hydroquinone 301 nm). All the experiments were performed in triplicate at room temperature, and the reported results are average values with standard deviation. All glassware was drenched in dilute nitric acid for 6 h and finally rinsed 3 times with distilled water prior to use.

*Influence of adsorbent dosage.* The effects of adsorbent dosage on the removal of metal ions and organic compounds were evaluated using different dosages of rGO (5, 10, 20, 30, and 40 mg). The initial concentration of metal ions (200 mg/L) and organic compounds (100 mg/L) were prepared by diluting standard stock solutions with deionized water at room temperature.

*Effects of pH.* The initial pH of the working solution was adjusted to the required value by adding 1M of NaOH or HNO_3_ solution before mixing with the adsorbents to study the effect of pH. For organic compounds, the solution pH was adjusted with solutions of 0.1 M NaOH and/or HCl. The adsorption measurements were similar to those used for the measurement of the previous batch mode adsorbent experiments.

*Effect of the adsorbate concentration.* The adsorption behavior for different concentrations of metal ions of Ag (I), Cu(II), Fe (II), Mn (II), and Pb(II) were studied in detail. The batch adsorption experiments were carried out by mixing rGO (10 mg for −30 mg) with different concentrations (5, 10, 25, 50, 100, 150, and 200 mg/L) of heavy metals in solution (20 mL). For methylene blue and hydroquinone, 10 mg of rGO was mixed with working solutions in test tubes.

The percentage removal and amount of metal ions and organic compounds adsorbed at equilibrium (q_e_) was calculated using Equations (1) and (2), respectively
(1)$$\mathrm{Removal}\;\left(\%\right)=\frac{C_{0}-C_{e}}{C_{0}}*100\;$$
(2)$$q_{e}=\frac{V\;\left(C_{O}-C_{e}\right)}{m}$$
where *C_0_* and *C_e_* are the initial and the equilibrium concentrations (mg/L), respectively, of the metal ions of each remaining solutions. *V* is the volume of the heavy metal ion solution, and *m* is the mass of the adsorbent.

*Langmuir isotherm.* The Langmuir isotherm was used to determine the maximum adsorption capacity produced from complete monolayer coverage of adsorbent surface, as shown in the isotherm Equation (3)
(3)$$q_{e}=\frac{q_{max}K_{L}C_{e}}{1+K_{L}C_{e}}$$ which on linearization of Equation (3) becomes
(4)$$\frac{C_{e}}{q_{e}}=\frac{1}{q_{max}K_{L}}+\frac{C_{e}}{q_{max}}$$
where *q_e_* depicts the concentration of metal ion or organic compound in the solution at equilibrium after sorption, *C_e_* is the equilibrium concentration of the adsorbate (mg/L), *K_L_* is the quotient of the rate of adsorption over the rate of desorption (L/mg), and *q_max_* is the Langmuir maximum adsorption monolayer capacity (mg/g) [77].

According to Equation (4), a plot of *C_e_*/*q_e_* versus *C_e_* should be a straight line with a slope 1/q_max_ and intercept 1/*q_max_K* when adsorption follows the Langmuir equation. The separation factor, *R_L_*, was obtained using Equation (5):(5)$$R_{L}=\frac{1}{1+C_{0}K}$$

The value of *R_L_* lies between 0 and 1 for a favorable adsorption. An *R_L_* > 1 represents an unfavorable adsorption, *R_L_* = 1 shows the linear adsorption, and the adsorption process is irreversible if *R_L_* = 0.

*Freundlich isotherm*. The Freundlich isotherm can be expressed as
(6)*q_e_ = K_f_ C_e_1/n*
where *K_f_* is the Freundlich constant, which gives the relative adsorption capacity of the adsorbent related to the bonding energy, and 1/*n* is the heterogeneity factor. On linearization of Equation (6), the plot of log *q*_e_ against log *C_e_* was used to determine the Freundlich coefficient as given below [77]
(7)$$\mathrm{Log}q_{e}=LogK_{f}+\frac{1}{n}Log\;C_{e}$$

*Adsorption of residues from paper photography processing.* The analysis of a real sample using wastewater from paper photograph processing was carried out to verify the practicability of the proposed method. The total dissolved solids (TDS) measurements were determined in the real sample before and after treatment with rGO, recorded with three repeatable readings using a Jenway 4510 conductivity/TDS meter (Jenway, Staffordshire, United Kingdom). Additionally, a rotary evaporator was used to remove water from 30 mL of the photo-processing sample. The same procedure was carried out using the samples after rGO treatment. All glassware was drenched in dilute nitric acid for 6 h and finally rinsed 3 times with distilled water prior to use.

## 4. Conclusions

In conclusion, we have presented a low-cost and simple synthesis of reduced graphene oxide, producing a material capable of adsorbing metal ions and organic contaminants. Our new method uses gallic and sodium ascorbate without any chemical functionalization or high thermal processing, making it accessible in terms of cost and processing requirements. Gallic acid and sodium ascorbate produce rGO with a high surface area density for metal ion adsorption. We demonstrated the competitive adsorption capacities of several ion metals, as well as of methylene blue and hydroquinone, highlighting a maximum absorption of 243 mg/g for Pb(II), 238 mg/g for MB, and 150 mg/g for HQ. We also demonstrated effective cleaning in a real sample, such as real wastewater from paper photography processing. A Langmuir model of the isotherms provides a good fitting for the adsorption of metal ions, blue methylene, and hydroquinine (at low concentrations), corresponding to a monolayer process. However, for hydroquinone at high concentrations, a Freundlich isotherms model provides a better fitting, indicating a multilayer adsorption mechanism. Compared to just reduced graphene oxide without functionalization, the adsorption of our materials has very strong performance. However, our adsorption capacity is still below those of functionalized graphene-based materials, which require additional synthesis and functionalization treatments. Further work will look into the time adsorption characteristics for all of these compounds and explore possible mechanisms and adsorption phases for hydroquinone at high concentrations. Our intention here is to show the effectiveness of the technique to adsorb water contaminants, and our future plan will be to understand adsorption kinetics. However, the results of this work show that an effective cleaning material can be produced starting from graphite without any heating treatment, which can be used by communities with limited resources facing water contamination by metal ions or organic compounds.

## Figures and Tables

**Figure 1 molecules-28-00649-f001:**
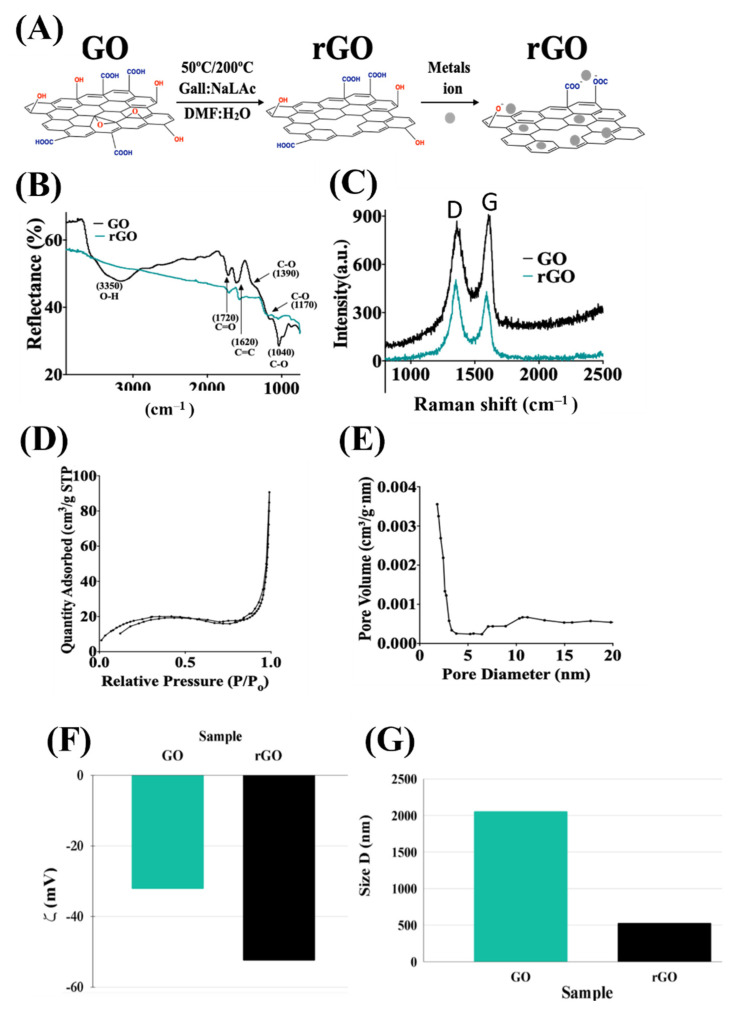
Characterization of rGO. (**A**) Reduction of GO into rGO and its interaction with metal ions. (**B**) FTIR spectra of GO and rGO, showing the conversion of GO to rGO as the peaks associated with oxygen functional groups in GO are removed in rGO spectra. (**C**) Raman analysis of GO and rGO, showing the increase in I_D_/I_G_ ratio (D: 1365 cm^−1^, G: 1595 cm^−1^), as expected for graphene reduction. (**D**) N_2_ adsorption and desorption isotherms. (**E**) Pore size distribution for rGO extracted from nitrogen isotherms. (**F**) Zeta potential values for GO and rGO, indicating a larger number of negative surface charges after reduction to rGO, resulting in strong attraction for positive metal ions. (**G**) Estimation of GO and rGO flake size from zeta-potential.

**Figure 2 molecules-28-00649-f002:**
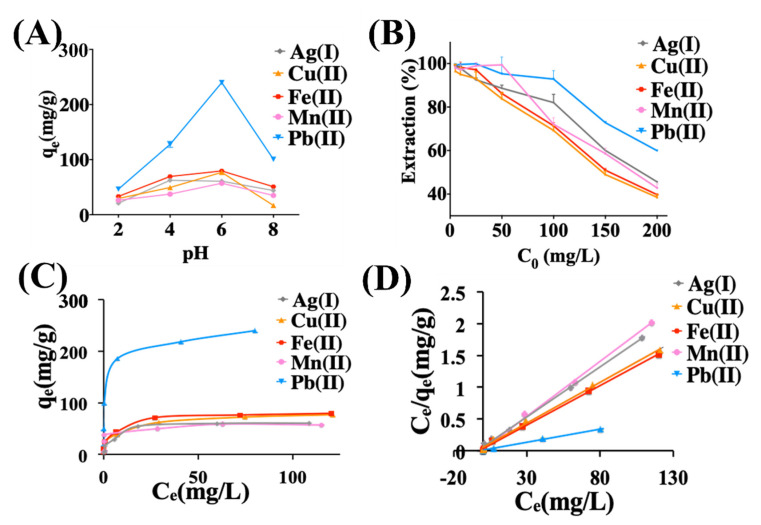
Adsorption of metal ions by rGO. (**A**) Effect of pH values on the adsorption capacity of Ag (I), Cu (II), Fe (II), Mn (II), and Pb (II), showing optimal adsorption at pH = 6. Metal ion concentration is 200 mg/L (ppm). (**B**) Effect of the initial concentration on the extraction of metal ions (C_0_ = 5–200 mg/L, ppm). (**C**) Plot of q_e_ versus C_e_ and (**D**) Experimental data and the fitted Langmuir isotherm based on the plot of C_e_/q_e_ versus C_e_. Experiment conditions. Adsorbent dose: Ag (I) = 30 mg/20 mL, Cu (II) = 20 mg/20 mL, Fe (II)=20 mg/20 mL, Mn (II) = 30 mg/20 mL, and Pb (II) = 10 mg/mL. Temperature = 25 ± 2 °C, Ph = 6, contact time = 48 h.

**Figure 3 molecules-28-00649-f003:**
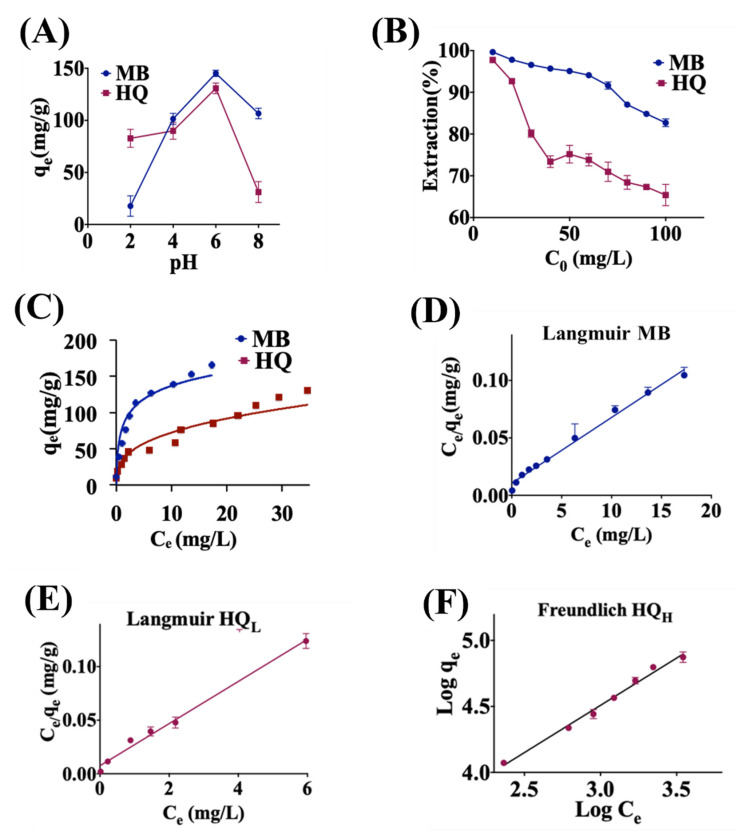
Adsorption isotherms of rGO for MB and HQ. (**A**) Adsorption capacity as a function of pH for MB and HQ. MB and HQ concentrations: 100 mg/L (ppm). (**B**) Extraction (%) as a function of C_0_ for MB and HQ. (**C**) Plot of q_e_ versus C_e_ for MB and HQ. (**D**) Fitted Langmuir isotherm for MB (C_e_/q_e_ vs. C_e_), from which a maximum capacity of *q*_max_ = 238.45 mg/g is extracted. (**E**) Fitted Langmuir isotherm based (C_e_/q_e_ vs. C_e_) for low concentration HQ (HQ_L_, 5–30 mg/L), from which a *q*_max_ = 51.02 mg/g is extracted. (**F**) Freundlich isotherm (Log q_e_ vs. Log C_e_) for high concentration HQ (HQ_H_, 40–100 mg/L), from which a K_F_ = 1.229 is extracted. Experiment conditions for MB and HQ: Adsorbent dose: 10 mg/20 mL. Temperature = 25 ± 2 °C, pH = 6, contact time = 48 h.

**Figure 4 molecules-28-00649-f004:**
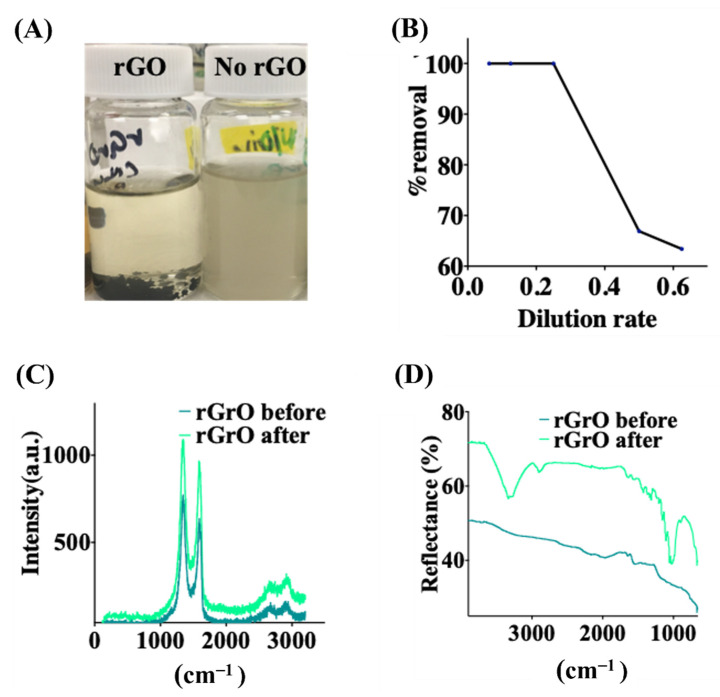
Treatment of wastewater from photo-processing operations (WPO) using rGO. (**A**) Photos for the treatment of WPO sample with rGO after 48 h and a control sample without rGO. (**B**) Percentage of adsorption (Removal%) of chemical compounds from WPO by rGO. (**C**) Raman analysis of rGO before and after adsorption, showing that the basal structure of graphene material remains intact. (**D**) FT-IR spectra of rGO before and after treatment of WPO, showing how a significant number of groups bind to graphene, indicating adsorption by rGO.

**Table 1 molecules-28-00649-t001:** Adsorption parameters of rGO for metal ions based on Langmuir isotherms. (Extracted from fitting of Figure 2D to Equation (4)).

Metal	*q*_max_ (mg/g)	K_L_ (mg/L)	R_L_	R^2^
Mn(II)	57.80	0.0232	0.1773	0.9984
Cu(II)	78.12	0.0505	0.0900	0.9965
Pb(II)	243.90	0.0086	0.3676	0.9942
Ag(I)	63.29	0.0593	0.0777	0.9970
Fe(II)	80.64	0.0302	0.1420	0.9983

**Table 2 molecules-28-00649-t002:** Comparison of adsorption capacity of metal ions (*q_max_*, mg/g) of rGO with other graphene adsorbents.

Material	Metal ion	*q*_max_ (mg/g)	Reference
RGO/Fe_3_O_4_ Magnetic Nanoparticles [86]	Pb (II)	30	[86]
RGO-Fe_3_O_4_ Hybrid Nanocomposite [97]	Pb (II)	50	[97]
Nickel Ferrite-Reduced Graphene Oxide Nanocomposite [98,99]	Pb (II)	120−150	[98,99]
Reduced Graphene Oxide aerogel [73]	Pb (II)	58.04	[73]
2-Imino-4-Thiobiuret-Partially Reduced Graphene Oxide (IT-PRGO) [70]	Pb (II)	102.2	[70]
	Cu (II)	37.9	
rGO from algal extracts [100]	Pb (II)	95	[100]
	Cu (II)	90	
rGO (with ethylenediamine) [71]	Pb (II)	413.34	[71]
	Cu (II)	55.34	
	Mn (II)	42.46	
**rGO by by Gallic Acid and Sodium Ascorbate**	Pb (II)	243.90	Present study
	Cu (II)	78.12	
	Mn (II)	57.83	
	Ag (I)	63.29	
	Fe (II)	80.64	
Chitosan/Graphene Oxide Composite [102]	Au (III)	1076.65	[102]
	Pd (II)	216.92	
Chitosan/Graphene Oxide Composite Nanofibrous [101]	Pb (II)	461.3	[101]
	Cu (II)	423.8	
	Cr (VI)	310	

**Table 3 molecules-28-00649-t003:** rGO adsorption parameters for MB and HQ.

Methylene Blue				
**Langmuir**	***q*_max_ (mg/g)**	**K_L_ (mg/L)**	**R_L_**	**R^2^**
MB	238.45	0.0164	0.3787	0.9878
**Freundlich**	**K_f_**	**1/n_F_**	**n_F_**	**R^2^**
MB	1.381	2.19	0.4556	0.973
**Hydroquinone at low (HQ_L_) and High (HQ_H_) C_e_**				
**Langmuir**	***q*_max_ (mg/g)**	**K_L_ (mg/L)**	**R_L_**	**R^2^**
HQ_L_	51.020	0.0067	0.5988	0.9937
HQ_H_	238.09	0.0959	0.0944	0.7235
**Freundlich**	**K_f_**	**1/n_F_**	**n_F_**	**R^2^**
HQ_L_	1.229	0.277	3.607	0.9780
HQ_H_	0.834	0.709	1.408	0.9923

**Table 4 molecules-28-00649-t004:** Comparison of adsorption capacity of MB (*q_max_*, mg/g) of rGO with other graphene adsorbents.

Adsorbent Material	q_max_ (mg/g)	Reference
Self-assembled graphene-carbon nanotube hybrid [118]	81.97	[118]
Eco-friendly rGO [119]	121.95	[119]
Pineapple peel carboxy methylcellulose-g-poly (acryliccid-co-acrylamide)/graphene oxide hydrogels [120]	133.32	[120]
rGO by hydrazine reduction of GO [121]	153.85	[121]
Alginate modified graphene [122]	159	[122]
Carboxymethyl cellulose/carboxylated graphene oxide composite microbeads [123]	180.23	[123]
Graphene oxide/calcium alginate composites [109]	181.81	[109]
Graphene nanoplatelets [108]	225	[108]
**rGO by Gallic Acid and Sodium Ascorbate**	**238.45**	**Present Study**
GO with high-oxidation degree [110,111]	600–1635	[110,111]

**Table 5 molecules-28-00649-t005:** Comparison of adsorption capacity of HQ (*q_max_*, mg/g) of rGO with other graphene adsorbents.

Adsorbent Material	q_max_ (mg/g)	Reference
Organobentonites (ODTMA-B, HDTMA-B) [112]	12.05–21.55	[112]
Iron (Fe) impregnated granular activated carbon (Fe-GAC) [113]	26.65	[113]
Graphene aerogels–mesoporous silica (GAs–MS) [83]	67	[83]
Granular activated carbon (GAC) [114]	102.3–135.3	[114]
**rGO by Gallic Acid and Sodium Ascorbate**	**~150**	**Present Study**
*Phragmites australis* activated carbon (PAAC) [115]	156.25	[115]
Cationic amino-poly(vinylamine) (PVAm)-functionalized GO-(o-MWCNTs)-Fe_3_O_4_ [116]	293.25	[116]
Magnetic cyclodextrin chitosan/graphene oxide (CCGO) [117]	428.72	[117]

## Data Availability

Data available upon request to authors.

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
