# Peer review of "Effective Removal of Metal ion and Organic Compounds by Non-Functionalized rGO"

_molecules, 2023, doi:10.3390/molecules28020649_

Round 1
Reviewer 1 Report
In this manuscript, Sarmiento et al. reports the formation of a non-functionalized rGO via a simple and low-cost method. The authors gave extensive comparisons with other studies and applied this rGO in cleaning real wastewater from paper photography processing, which demonstrate that their technique is an effective way to adsorb metal ions and organic contaminants. The work is intact and interesting, and it can be considered for the publication in this journal after revising the following few issues:
1. In Equation (6), should be nF.
2. In Figure 2A, for Pb (II), qe is over 200 mg/g at pH=6, but in Line 237, the authors described “…reaching 190 mg/g at pH=6,”.
3. In reference [90], the prepared chitosan/graphene oxide composites were used for the adsorption of Au(III) and Pd(II). Table 2 in this manuscript gives the wrong information.
Moreover, Line 272-273: the authors described “Our performance is still significant lower than functionalized GO-based materials that can reach qmax~500 mg/g using chitosan/GO90, or chitosan/GO nanofibers.91” This description is not accurate.
4. In Table 3, the Freundlich model with high concentrations shows Kf=0.834, but in Line 314: “…of Kf=1.229.”
5. In Page 13, should be Figure 4.
Reviewer 2 Report
I have read the manuscript thoroughly entitled ‘
Effective removal of metal ion and organic compounds by non- 2 functionalized rGO. I have reached the conclusion that the author made effort to do the citable study but at the same time manuscript need some changes.
English need to refine through the manuscript. After these all-major corrections, manuscript can be sent for decision by the editor. Citation is not even appropriate.
My recommendation is to accept this manuscript after these change.
The abstract must be more descriptive in a concise way. The first part of the abstract not needed because it is described in the introduction. Also in the characterization part, there must be SEM to show the correct morphology of the prepared and before reduced, and after reduction.
Reviewer 3 Report
I have reviewed entitled "Effective removal of metal ion and organic compounds by non-functionalized rGO ". I would like to recommend the acceptance of this manuscript with major corrections required.
The details of the comments are listed below:
1. What are the differences between this study and others in the literature? The originality/novelty of the paper should be clearly stated in the manuscript.
2. The English throughout the manuscript must be improved and revised.
3. In the introduction section, it is necessary to write materials about, the effects of metal ions and organic compounds on the environment, and the removal methods of pollutants. It is suggested that the introduction section be improved using the suggested articles:
-https://doi.org/10.1007/s13201-015-0322-y
-https://doi.org/10.1007/s10924-021-02188-1
-https://doi.org/10.1016/j.clay.2021.106405
-https://doi.org/10.1016/j.gsd.2022.100728
- https://doi.org/10.1016/j.emcon.2022.01.002
-https://doi.org/10.1016/j.matpr.2022.08.412
-https://doi.org/10.1007/s41204-021-00173-6
-https://doi.org/10.1002/jctb.7183
-https://doi.org/10.1007/s13369-022-06580-4
4. Surface area of the adsorbent is an important characteristic (BET). Please provide the surface area, total pore volume, and average pore size of the prepared adsorbent (rGO).
5. What was the mechanism of action of binding of metal ions and organic compounds onto the surface of rGO
6. Conclusion also needs to be rewritten. Include the following: new concepts and innovations demonstrated in this study, a summary of findings, a comparison with findings by other workers, and a concluding remark
Round 2
Reviewer 3 Report
Accept